# ROR1 as an Immunotherapeutic Target for Inducing Antitumor Helper T Cell Responses Against Head and Neck Squamous Cell Carcinoma

**DOI:** 10.3390/cancers17142326

**Published:** 2025-07-12

**Authors:** Ryosuke Sato, Hidekiyo Yamaki, Takahiro Inoue, Shota Sakaue, Hisataka Ominato, Risa Wakisaka, Hiroki Komatsuda, Michihisa Kono, Kenzo Ohara, Akemi Kosaka, Takayuki Ohkuri, Toshihiro Nagato, Takumi Kumai, Kan Kishibe, Hiroya Kobayashi, Miki Takahara

**Affiliations:** 1Department of Otolaryngology-Head and Neck Surgery, Asahikawa Medical University, Asahikawa 078-8510, Japan; rsato@asahikawa-med.ac.jp (R.S.); hidekiyo@asahikawa-med.ac.jp (H.Y.); inotaka9242@asahikawa-med.ac.jp (T.I.); syouta0701@asahikawa-med.ac.jp (S.S.); h-ominato@asahikawa-med.ac.jp (H.O.); r-wakisaka@asahikawa-med.ac.jp (R.W.); komatsuda@asahikawa-med.ac.jp (H.K.); mkono@asahikawa-med.ac.jp (M.K.); kenzo@asahikawa-med.ac.jp (K.O.); kkisibe@asahikawa-med.ac.jp (K.K.); miki@asahikawa-med.ac.jp (M.T.); 2Department of Innovative Head & Neck Cancer Research and Treatment, Asahikawa Medical University, Asahikawa 078-8510, Japan; 3Department of Pathology, Asahikawa Medical University, Asahikawa 078-8510, Japan; kosakaa@asahikawa-med.ac.jp (A.K.); ohkurit@asahikawa-med.ac.jp (T.O.); rijun@asahikawa-med.ac.jp (T.N.); hiroya@asahikawa-med.ac.jp (H.K.)

**Keywords:** ROR1, peptide vaccine, helper T cells, head and neck squamous cell carcinoma, immune-checkpoint inhibitors, PD-1, PD-L1, PD-L2

## Abstract

ROR1, a tumor-associated antigen (TAA), is widely expressed in various cancers. However, its expression in HNSCC remains poorly characterized. Given the demonstrated tolerability of ROR1-targeting therapies in clinical trials, ROR1 may represent a promising TAA for T cell-based peptide vaccine development. Here, we demonstrate that ROR1 is widely expressed in HNSCC tissue specimens and cell lines, with significantly higher expression in HNSCC than in healthy tissues. To develop an ROR1-targeted peptide vaccine, we identified a novel ROR1-derived epitope capable of eliciting antitumor T cell responses against HNSCC. ROR1-reactive helper T cell (HTL) lines secreted effector cytokines and exhibited direct cytotoxic activity against ROR1+ HNSCC cell lines in a human leukocyte antigen (HLA)-DR-restricted manner. Furthermore, the tumoricidal activity of T cells was enhanced by ICIs targeting the PD-L1/PD-1 and PD-L2/PD-1 axes. These findings suggest that ROR1 could serve as a promising immunotherapeutic target in patients with HNSCC.

## 1. Introduction

Head and neck squamous cell carcinoma (HNSCC) is the seventh most common cancer worldwide, with approximately 890,000 new cases and 450,000 deaths annually, according to Global Cancer Statistics [1]. More than half of patients with HNSCC are diagnosed at an advanced stage. Although multimodal treatments—including surgery, chemotherapy, and radiotherapy—are employed for HNSCC, treatment resistance frequently occurs, particularly in advanced cases [2]. In recent years, immunotherapies have garnered increasing attention, with immune checkpoint inhibitors (ICIs) now approved for patients with recurrent or metastatic HNSCC [3,4]. However, only approximately 15% of patients respond clinically to ICIs [3]. A major limitation of ICIs is their nonspecific activation of T cells. Rather than selectively targeting tumor-specific T cells, ICIs may also activate autoreactive T cells, leading to immune-related adverse events. Cancer vaccines offer a more targeted form of immunotherapy by selectively stimulating tumor-reactive T cells [5]. Although early clinical trials have produced disappointing results owing to suboptimal adjuvants, recent studies have accumulated supportive evidence for the efficacy of cancer vaccines [6].

Receptor tyrosine kinase-like orphan receptor 1 (ROR1), a tumor-associated antigen (TAA), is widely expressed in various cancers [7]. However, its expression in HNSCC remains poorly characterized. Given the demonstrated tolerability of ROR1-targeting therapies in clinical trials using chimeric antigen receptor T cells [8,9], ROR1 may represent a promising TAA for T cell-based peptide vaccine development. In this study, we demonstrate that ROR1 is widely expressed in HNSCC tissue specimens and cell lines, with significantly higher expression in HNSCC than in healthy tissues. To develop an ROR1-targeted peptide vaccine, we identified a novel ROR1-derived epitope capable of eliciting antitumor T cell responses against HNSCC. ROR1-reactive helper T cell (HTL) lines secreted effector cytokines and exhibited direct cytotoxic activity against ROR1+ HNSCC cell lines in a human leukocyte antigen (HLA)-DR-restricted manner. Furthermore, the tumoricidal activity of T cells was enhanced by ICIs targeting the PD-L1/PD-1 and PD-L2/PD-1 axes. These findings suggest that ROR1 could serve as a promising immunotherapeutic target in patients with HNSCC.

## 2. Materials and Methods

### 2.1. Patients and Immunohistochemistry

Tissue samples were obtained from pretreatment biopsy specimens of 30 patients with HNSCC treated at Asahikawa Medical University. TNM staging was performed according to the 8th edition of the Union for International Cancer Control guidelines. ROR1 expression was evaluated using formalin-fixed, paraffin-embedded tissues. A rabbit polyclonal antibody against ROR1 (1:400 dilution; Invitrogen, Waltham, MA, USA, RRID: AB_2182472) was used as the primary antibody. Immunohistochemistry (IHC) was performed using the ImmPRESS^®^ HRP Universal PLUS Polymer Kit, Peroxidase (Vector Laboratories, Newark, CA, USA). The staining intensity of ROR1 in tumor cells was graded on a four-point scale: 0 (negative), 1 (weak), 2 (moderate), and 3 (strong). The quantity score was based on the percentage of positively stained tumor cells and categorized as follows: 0 (<5%), 1 (5–25%), 2 (26–50%), and 3 (51–100%). The overall IHC score was calculated as the sum of the staining intensity and quantity scores. ROR1 expression was then categorized as negative (scores 1–2), low (3–4), or high (5–6). Informed consent was obtained through an opt-out method via the Asahikawa Medical University website (https://www.asahikawa-med.ac.jp/bureau/shomu/rinri/koukai.html, 12 July 2025). Clinical data collection and analysis was approved by the Institutional Review Board of Asahikawa Medical University (approval no. 23070). ROR1 expression in HNSCC cell lines was also evaluated by immunocytochemistry and Western blot. The cell lines (1 × 10^5^ cells) were cultured on glass slides overnight, fixed with 4% formalin, and permeabilized with 0.1% Tween-20. Staining was then performed following the same protocol as described above. In Western blot, Polyclonal rabbit anti-human ROR1 Ab (1:250; Santa Cruz Biotechnology, Santa Cruz, CA, USA) and monoclonal mouse anti-β-actin Ab (C4, 1:1000; Santa Cruz Biotechnology, Santa Cruz, CA, USA) were used.

### 2.2. Cell Lines

The following HNSCC cell lines were used: HPC92Y (hypopharyngeal SCC; HLA-DR4, -DR9, and -DR53), Sa3 (human gingival SCC; HLA-DR9, -DR10, and -DR53), and HSC3 (tongue SCC; HLA-DR15). Sa3 and HSC3 cells were obtained from the RIKEN BioResource Center. HPC92Y cells were kindly provided by Dr. Syunsuke Yanoma (Yokohama Tsurugamine Hospital, Japan). Mouse fibroblast cell lines (L-cells) expressing individual human HLA-DR molecules (DR4, DR15, or DR53) were kindly provided by Dr. T. Sasazuki (Kyushu University, Japan) and Dr. Robert W. Karr (Karr Pharma, St. Louis, MO, USA). All cell lines were cultured under conditions recommended by their respective suppliers.

### 2.3. Flow Cytometry

The expression levels of ROR1, PD-L1, and PD-L2 in the tumor cell lines were evaluated via flow cytometry using PE-conjugated antibodies: anti-ROR1 (1:200 dilution; 2A2, BioLegend, San Diego, CA, USA, RRID: AB_2561989), anti-PD-L1 (1:200 dilution; MIH1, eBioscience, San Diego, CA, USA, RRID: AB_11042286), and anti-PD-L2 (1:200 dilution; IPI683.rMab, BD Pharmingen, Heidelberg, Germany, RRID: AB_3684541). To assess the specificity of antibody staining, PE-conjugated mouse IgG1, κ isotype ctrl (1:200 dilution; MOPC-21, BioLegend, San Diego, CA, USA, RRID: AB_2847829) was used. Samples were analyzed using a CytoFLEX LX flow cytometer and CytExpert software (ver. 2.6, Beckman Coulter, Bres, CA, USA).

### 2.4. Synthetic Peptides

HLA-DR-binding amino acid sequences of ROR1 were identified using computational prediction algorithms from SYFPEITHI (http://www.syfpeithi.de/ accessed on 1 June 2024) and the Immune Epitope Database Analysis Resource (https://www.iedb.org/ accessed on 1 June 2024). These tools evaluated the peptide binding potential of common HLA-DR molecules (DR1, DR4, and DR7), leading to the selection of ROR1_403–417_ (EILYILVPSVAIPLA). The ROR1_403–417_ peptide was synthesized by Hokkaido System Science (Sapporo-shi, Japan).

### 2.5. In Vitro Induction of ROR1-Reactive CD4^+^ Helper T Cells

The procedure for generating ROR1_403–417_ peptide-reactive CD4+ HTLs from peripheral blood mononuclear cells (PBMCs) of healthy donors has been previously described in detail [10]. In brief, CD4+ HTLs were purified using the EasySep™ Human CD4-Positive T Cell Isolation Kit (STEMCELL Technologies, Vancouver, BC, Canada) and stimulated with peptide-pulsed autologous dendritic cells (DCs). The DCs were derived from CD14+ cells, which were isolated using the EasySep™ Human CD14-Positive Selection Kit (STEMCELL Technologies, Vancouver, BC, Canada) and subsequently stimulated with GM-CSF (50 ng/mL, PeproTech, Cranbury, NJ, USA) and IL-4 (1000 IU/mL, PeproTech). Following two rounds of stimulation with 40 Gy γ-irradiated autologous PBMCs (γPBMCs) and ROR1_403–417_ peptide, HTLs were assessed for IFN-γ production in response to peptide stimulation using Human IFN-γ ELISA kits (BD Pharmingen, La Jolla, CA, USA), according to the manufacturer’s protocol. Positive microcultures that exhibited a significant increase in IFN-γ production compared to unstimulated controls were then expanded. AIM-V medium (Thermo Fisher Scientific, Waltham, MA, USA) supplemented with 3% human male AB serum (Innovative Research, Novi, MI, USA) was used as the complete culture medium for all experiments.

### 2.6. Analysis of ROR1-Specific Responses with Established CD4+ Helper T Cell Lines

Established CD4+ HTLs (1–1.5 × 10^5^) were co-cultured with antigen-presenting cells (APCs)—including γPBMCs (1–1.5 × 10^5^) and HLA-DR-expressing L-cells (3 × 10^4^)—in the presence or absence of varying concentrations of the ROR1_403–417_ peptide in 96-well culture plates. Antigen specificity and HLA-DR restriction were confirmed by blocking antigen presentation with the anti-HLA-DR monoclonal antibody (L243, BioLegend, San Diego, CA, USA, RRID: AB_2800796) or anti-HLA class I monoclonal antibody (W6/32, BioLegend, San Diego, CA, USA, RRID: AB_2783175), both at 10 μg/mL, during a 48 h incubation. Supernatants were collected and analyzed for IFN-γ production using ELISA.

### 2.7. Cytotoxicity Assay

To investigate the anti-tumor activity of ROR1-reactive HTLs against HNSCC, HTLs were co-cultured with ROR1+ tumor cell lines (3 × 10^4^) for 48 h. Supernatants were collected and analyzed using Human IFN-γ and Granzyme B ELISA kits (R&D Systems, Minneapolis, MN, USA). To enhance HLA-DR expression, tumor cell lines were pretreated with IFN-γ (500 IU/mL) for 48 h prior to co-culture and thoroughly washed out before further analysis. The HTLs and tumor cell lines were co-cultured with anti–PD-1, PD-L1, or PD-L2 antibodies (10 μg/mL) for 48 h to assess the impact of immune checkpoint blockade on HTL-mediated anti-tumor activity. For direct cytotoxic assessment of ROR1-reactive HTLs, target tumor cell lines were labeled with the CellTrace™ CFSE Cell Proliferation Kit (1:20,000 dilution; Invitrogen, Carlsbad, CA, USA, RRID: not available). After a 6 h co-culture at various effector-to-target ratios between HTLs and tumor cells, dead tumor cells were quantified via flow cytometry using a 7-AAD (undiluted; BD Pharmingen, Heidelberg, Germany, RRID: AB_2869266) viability staining solution.

### 2.8. ROR1_403–417_ Peptide-Reactive Responses in Patients with HNSCC and Healthy Individuals

PBMCs from patients with HNSCC and healthy individuals were co-cultured with ROR1_403–417_ peptides in 96-well plates following a previously described protocol [10]. Given the limited availability of PBMCs from patients, a short-term stimulation approach was employed. PBMCs were stimulated with the ROR1_403–417_ peptide in two cycles 7 d apart. IFN-γ and granzyme B production in the culture supernatants was measured using ELISA. An anti-DR antibody was used to assess HLA-DR restriction. All experiments were conducted with the approval of the Institutional Ethics Committee of Asahikawa Medical University (approval no. 23070), and written informed consent was obtained from all participants.

### 2.9. Statistical Analysis

Patient characteristics were compared between the high ROR1 and low/negative expression groups using Fisher’s exact test. Differences in ROR1 expression between HNSCC tissues and control tonsillar epithelia were analyzed using the Mann–Whitney U test. Progression-free survival (PFS) and overall survival (OS) were compared between the high and low/negative ROR1 expression groups using Kaplan–Meier survival curves and log-rank tests. All other statistical comparisons were performed using Student’s *t*-test or one-way ANOVA. Data are presented as mean ± standard deviation. Statistical significance was defined as * *p* < 0.05, ** *p* < 0.01, *** *p* < 0.001, and **** *p* < 0.0001. All statistical analyses were performed using GraphPad Prism (version 10.4.2, GraphPad Software Inc., La Jolla, CA, USA).

## 3. Results

### 3.1. ROR1 Expression in HNSCC Tissues and Cell Lines

We performed IHC staining to evaluate ROR1 expression in tissue samples from 30 patients with HNSCC and 6 healthy tonsillar epithelia. ROR1 expression in HNSCC patients and healthy controls is shown in Figure 1a–d. Among the healthy controls, five of six (83.3%) showed negative, while one (16.7%) showed low ROR1 expression (Figure 1a,c). Among the 30 patients with HNSCC, 6 (20.0%) were negative, 12 (40.0%) exhibited low expression, and 12 (40.0%) had high ROR1 expression (Figure 1b,c). The median IHC scores for ROR1 were 4.0 in patients with HNSCC and 2.0 in healthy controls. ROR1 IHC scores were significantly higher in HNSCC tissues than in healthy controls (Figure 1e).

Table 1 summarizes the clinical characteristics of patients stratified by high or low/negative ROR1 expression. High ROR1 expression was significantly associated with T3–4 tumors, N2–3 nodal status, and advanced clinical stage. To evaluate the relationship between ROR1 expression and HNSCC progression, PFS and OS were compared between patients with high versus low/negative ROR1 expression. As shown in Figure 1d,e the 5-year PFS rate was 55.6% in the high-expression group and 62.7% in the low/negative group (*p* = 0.75). The 5-year OS rate was 51.6% in the high-expression group and 68.6% in the low/negative group (*p* = 0.36). Although ROR1 was expressed in 80% of HNSCC cases and correlated with advanced stage, PFS and OS were comparable between patients with high and low/negative expression.

### 3.2. Generation of ROR1_403–417_—Reactive HTLs

Given that most HNSCC tissues express ROR1 in clinical samples, we examined ROR1 expression in HNSCC cell lines. The positive expression of ROR1 was confirmed in multiple HNSCC cell lines (HPC92Y, Sa3, and HSC-3) using flow cytometry (Figure 2a). ROR1 positivity was also confirmed in these cell lines by immunocytochemistry and Western blot (Appendix A). No apparent differences in expression were observed among the tumor cells. To develop ROR1-targeted T cell immunotherapy, we employed computational T cell epitope prediction tools and identified the ROR1_403–417_ peptide (EILYILVPSVAIPLA) as a candidate peptide capable of eliciting CD4^+^ HTL responses. Purified CD4^+^ T cells from healthy donors were repeatedly stimulated with ROR1_403–417_ peptide-loaded APCs. As a result, two ROR1_403–417_-reactive HTL lines, designated R1 and R2, were successfully established. These HTLs secreted IFN-γ in response to ROR1_403–417_ stimulation in a dose-dependent manner (Figure 2b). T-cell responses were blocked using an anti-HLA-DR antibody but not with anti-HLA class I antibodies, indicating that antigen recognition was restricted to HLA-DR (Figure 2c). To identify the specific HLA-DR allele, L cells expressing individual HLA-DR molecules were used as APCs. The HTLs responded specifically to L cells expressing HLA-DR53 (Figure 2d), a molecule associated with the common HLA-DR alleles (DR4, DR7, and DR9). These findings indicate that the ROR1_403–417_ epitope is immunogenic and potentially applicable to a broad patient population.

### 3.3. Direct Tumor Recognition and Cytokine Production by ROR1-Reactive HTLs

Next, we investigated whether ROR1-reactive HTLs exhibit antitumor activity against ROR1-expressing tumor cell lines. As shown in Figure 3a, these HTLs secreted IFN-γ in response to HLA-DR53-expressing HPC92Y and Sa3 cells. This response was significantly reduced in the presence of an anti-HLA-DR antibody. In contrast, no reactivity was observed against HLA-DR-unmatched ROR1+ HSC3 cells, suggesting that T cells reacted to the tumor in the context of the HLA-DR–peptide–T cell receptor interaction. To further evaluate functional activity, we measured granzyme B production by the ROR1-reactive HTLs (Figure 3b). Both HTL lines secreted granzyme B in response to HLA-DR-matched tumor cell lines, but not in HLA-DR-mismatched lines. To assess direct cytotoxicity, CFSE-labeled tumor cells were co-cultured with the HTLs. The representative example of flow gating was shown in Appendix A. The ROR1-reactive HTLs effectively induced direct cytotoxicity against HLA-DR-matched tumor cells (Figure 3c,d). These results suggest that the ROR1_403–417_ peptide could serve as a promising candidate for peptide-based cancer vaccines.

### 3.4. Antitumor Activity of ROR1-Reactive HTLs Enhanced by PD-L1/PD-1 and PD-L2/PD-1 Axes Blockade

The negative immune checkpoint ligands PD-L1 and PD-L2 expressed on tumor cells suppress antitumor immune responses by engaging PD-1 on immune cells. Clinical trials have demonstrated that PD-1 inhibitors improve survival in patients with HNSCC [2]. Given that the activation of T cells through T cell receptor signaling induces PD-1 expression [11], PD-1 blockade may be an effective adjuvant in peptide vaccines. To assess whether the blockade of PD-L1/PD-1 and PD-L2/PD-1 axes enhances the antitumor activity of ROR1-reactive HTLs, HTLs were co-cultured with HLA-DR-matched HPC92Y cells in the presence of ICIs targeting PD-1, PD-L1, PD-L2, or both PD-L1 and PD-L2. HPC92Y cells expressed both PD-L1 and PD-L2, and their expression was further upregulated following IFN-γ treatment (Figure 4b). As shown in Figure 4c, all types of ICIs enhanced IFN-γ production. Notably, PD-1 and combined PD-L1/PD-L2 blockade induced significantly higher IFN-γ and granzyme B levels than either PD-L1 or PD-L2 blockade alone (Figure 4c,d). These findings suggest that the dual blockade of the PD-L1/PD-1 and PD-L2/PD-1 axes is important for enhancing the anti-tumor efficacy of HTL peptide vaccines against ROR1 in HNSCC.

### 3.5. Detection of ROR1_403–417_—Reactive T Cells in PBMCs from Patients with HNSCC and Healthy Individuals

The presence of peptide-reactive precursor T cells is essential for eliciting effective antitumor responses in peptide vaccines. To evaluate the existence of ROR1-reactive T cells in circulating immune cells, PBMCs from patients with HNSCC and healthy individuals were stimulated with the ROR1_403–417_ peptide. Since the frequency of ROR1-reactive T cells was low (Appendix A), we employed a long-term ex vivo expansion strategy using limited sample volumes. The clinical characteristics of the four patients and four healthy individuals are summarized in Table 2. All patients with HNSCC exhibited detectable T cell responses to ROR1_403–417_, which were inhibited upon the addition of the anti-HLA-DR antibody (Figure 5a). In contrast, two of the four healthy individuals exhibited a measurable response to the peptide (Figure 5b). These findings suggest that ROR1-reactive precursor T-cells are present in patients with HNSCC and suggest their potential utility in the development of ROR-targeted peptide vaccines.

## 4. Discussion

Optimal targets for cancer vaccines are characterized by high expression in malignant cells and minimal or absent expression in healthy tissues. Neoantigens—tumor-specific peptides arising from somatic mutations—possess high immunogenicity [12]. Neoantigen-based immunotherapies have shown promising clinical efficacy in several malignancies [13,14,15]. However, such immunotherapies present several challenges. The identification and validation of immunogenic, non-shared neoantigens are time-consuming and expensive. Moreover, their therapeutic efficacy can be compromised by secondary mutations within the neoantigen sequence, limiting their broader applicability [5]. To address these limitations and expand the clinical applicability of peptide vaccines, we focused on targeting TAAs that are linked to dysregulated proliferation pathways in tumor cells yet exhibit minimal expression in healthy cells. Among these, tumor-associated fetal antigens are particularly promising for immunotherapy. Fetal antigens are typically expressed during embryonic development but are aberrantly re-expressed in cancer cells, with minimal expression in healthy adult tissues [16].

ROR1 is a fetal antigen that belongs to the tyrosine kinase receptor superfamily and plays a critical role in the early development of the nervous system and skeletal tissues. It is overexpressed in various hematologic malignancies and solid tumors [17], whereas its expression is low or undetectable in most normal adult tissues. One study reported positive ROR1 expression in specific normal adult tissues, including the parathyroid gland, pancreas, esophagus, stomach, and duodenum [7]. However, other studies found no ROR1 expression in a broad range of normal adult tissues, including immune-related organs (bone marrow, peripheral blood immune cells, spleen, thymus, tonsil, and lymph nodes), heart, artery, brain, lung, liver, pancreas, gastrointestinal tract, skeletal muscle, kidney, prostate, testis, ovary, and uterus [7,18,19,20,21], suggesting that ROR1-targeted therapies have low risk to induce off-target adverse events.

Nevertheless, ROR1 expression in HNSCC has not been well characterized. In our study, ROR1 was significantly overexpressed in HNSCC tissues compared to normal tonsillar epithelium, with approximately 80% of HNSCC cases demonstrating positive ROR1 expression. This expression frequency is comparable to that reported in hematologic cancers [22,23,24], further supporting the rationale for targeting ROR1 in cancer immunotherapy with minimal immune-related adverse effects. Previous studies have shown that ROR1 contributes to cancer cell proliferation, migration, and angiogenesis [17]. In HNSCC, siRNA-mediated ROR1 knockdown inhibited oral cancer cell proliferation [25]. Moreover, a meta-analysis of multiple patients with cancer identified that high ROR1 expression correlated with poor prognosis [26]. Another study of oral cancer demonstrated that high ROR1 expression was associated with lymph node metastasis [27]. Consistent with these findings, our study demonstrated a significant association between high ROR1 expression and advanced T, N, and clinical stages in HNSCC. These findings suggest that ROR1-targeted immunotherapy may offer particular benefit to patients with advanced HNSCC.

Anti-ROR1 therapies have shown encouraging antitumor efficacy in several malignancies. Zilovertamab vedotin, an antibody–drug conjugate (ADC) targeting ROR1, demonstrated response rates of 47% and 60% in mantle cell and diffuse large B-cell lymphomas, respectively [28]. Another ROR1-targeted ADC, CS5001, achieved a 43.5% response rate in patients with advanced lymphomas [29]. In chronic lymphocytic leukemia, ROR1-specific chimeric antigen receptor T cell therapy elicited rapid antitumor responses in two of three treated patients [9]. However, evidence supporting the efficacy of ROR1-targeted therapies in solid tumors remains limited [9,29], and no studies have yet investigated their therapeutic potential in HNSCC. To the best of our knowledge, our study is the first to demonstrate the feasibility and efficacy of targeting ROR1 as a T-cell-based immunotherapy in HNSCC. The ROR1-reactive HTLs selectively recognized ROR1^+^ HNSCC cell lines in an HLA-DR-restricted manner, indicating that the ROR1_403–417_ epitope is naturally processed and presented by tumor cells. Furthermore, our results suggest that this epitope is a viable candidate for a peptide vaccine. In addition, our short-term T-cell activation protocol using the ROR1 peptide effectively identified tumor-reactive T-cells in PBMCs.

Traditionally, cancer vaccines have focused on eliciting CD8^+^ cytotoxic T lymphocytes (CTLs), which directly kill tumor cells. However, the clinical efficacy of CD8^+^ T cell-based vaccines was found to be unsatisfactory in trials during the 1990s. CD4^+^ HTLs were historically considered as indirect supportive cells that enhanced antitumor immunity by producing cytokines and co-stimulatory signals to CD8^+^ CTLs. More recently, however, accumulating evidence has demonstrated that CD4+ HTLs also possess direct cytotoxicity against tumor cells [30,31]. In our study, ROR1-reactive HTLs secreted IFN-γ in response to both the ROR1_403–417_ peptide and ROR1+ HNSCC cell lines. IFN-γ is a key cytokine that plays a crucial role in antitumor immunity by upregulating HLA molecules and activating multiple immune subsets, including CTLs, natural killer cells, DCs, and macrophages [32]. While IFN-γ can also induce PD-L1 and PD-L2 expression in tumor cells, our data showed that the combination of ICIs with peptide vaccination could counteract the negative effects of IFN-γ. In addition, ROR1-reactive HTLs produce granzyme B, which directly induces tumor cell death [33], as confirmed by their cytotoxicity against ROR1+ HNSCC cell lines. These findings suggest that HTL-targeted cancer vaccines may serve as an effective immunotherapeutic strategy by exerting not only supportive roles but also direct tumoricidal activity.

In recent years, the combination of cancer vaccines with ICIs has shown promising anti-tumor efficacy. In HNSCC, combination therapy with a human papillomavirus (HPV) type 16 vaccine and nivolumab achieved a favorable response rate of 33% [34]. Another study using a DNA vaccine targeting HPV-16/18 E6/E7 antigens in combination with durvalumab reported a response rate of 27.6% in patients with HPV-16+ HNSCC [35]. In our study, we demonstrated that the antitumor efficacy of ROR1-reactive HTLs was enhanced by the addition of ICIs. In melanoma, the combined use of cancer vaccines and ICIs has been evaluated in large-scale clinical trials; however, their therapeutic efficacy remains inconclusive. In a phase III trial, OS was significantly prolonged in patients treated with ipilimumab combined with a TAA-targeted peptide vaccine (gp100) compared to those receiving the vaccine alone (10.0 vs. 6.4 months) [36]. Conversely, another study reported no additional benefit of the gp100 vaccine when combined with ipilimumab, relative to ipilimumab monotherapy [37]. More recently, a clinical trial assessing an RNA-based neoantigen vaccine in combination with pembrolizumab as adjuvant therapy demonstrated a significant improvement in PFS compared to pembrolizumab alone [38]. These findings suggest that, in melanoma, ICIs are more likely to exert their effects via neoantigens rather than TAAs [39]. A limitation of neoantigen-based approaches is the potential loss of neoantigens, particularly in tumors with a high mutational burden, which may facilitate immune evasion. Furthermore, the identification of neoantigens is both costly and time-consuming. Although TAA-specific T cells generally exhibit lower functional potency compared to neoantigen-specific T cells, the therapeutic potential of TAA-specific T cells, especially in combination with ICIs, remains a compelling area for further investigation. Given the high mutational burden characteristic of melanoma, further research is warranted to assess whether TAA-based vaccine strategies combined with ICIs could also be effective in HNSCC.

In clinical practice, PD-1 inhibitors are the standard-of-care treatment for patients with HNSCC [2,3,4], while PD-L1 inhibitors are approved for use in other malignancies [40]. In addition to PD-L1, PD-L2 causes immunosuppression through interaction with PD-1 [41]. High PD-L2 expression has been associated with poor prognosis in HNSCC [42], and its expression has been reported to correlate with the efficacy of PD-1 inhibitors [43,44]. These findings suggest that the simultaneous blockade of both PD-L1 and PD-L2 may further enhance T cell responses compared to blockade of either ligand alone. As a proof of concept, our study demonstrated that the dual blockade of the PD-L1/PD-1 and PD-L2/PD-1 axes enhanced the anti-tumor efficacy of ROR1-reactive T cells compared to monotherapies. These findings highlight the importance of the PD-L2/PD-1 axis, besides PD-L1, in regulating antitumor immunity and support its consideration in future immunotherapeutic strategies.

A limitation of TAAs as targets for immunotherapy is their basal expression in normal cells. Most self-antigen-reactive T cells bearing high-affinity T cell receptors are eliminated from the thymus through negative selection. However, self-antigen-reactive T cells with low-affinity TCRs that can only react to tumor cells with high TAA expression survive negative selection [5]. In our study, we identified ROR1-reactive precursor T cells in both healthy donors and patients with HNSCC, none of whom had autoimmune pathology. These findings suggest that the precursor of ROR1-reactive T cells in patients with HNSCC can discriminate between tumor and normal tissues based on antigen expression thresholds. Furthermore, the tolerability of ROR1-targeted therapy in clinical trials supports the safety of pursuing ROR1 as a TAA in peptide vaccine strategies. In the clinical trial using the ROR1-targeted antibody–drug conjugate zilovertamab vedotin, the incidence of adverse events was consistent with that reported for other monomethyl auristatin E-containing antibody–drug conjugates [28]. Additionally, the safety of ROR1-targeted CAR-T cells has been reported [9]. Nevertheless, the safety and tolerability of ROR1-based peptide vaccines should be rigorously evaluated in future studies with in vivo studies or early-phase clinical trials.

The present study has several limitations. Firstly, the scope of our analysis was confined to in vitro experiments. As the safety and efficacy of peptide vaccine using other types of TAAs has been demonstrated in both animal and clinical trials [45], and we previously reported that a peptide vaccine targeting c-MET exhibited anti-tumor effects in an HNSCC mouse model [46], we believe that ROR1-targeted immunotherapy may offer clinical benefit in patients with HNSCC. Second, while high ROR1 expression appeared to be associated with poorer PFS and OS, the observed differences did not achieve statistical significance. Because high ROR1 expression was associated with advanced clinical stage, T classification, and N classification, we believe that the lack of a statistically significant association between ROR1 and survival is likely due to the relatively small sample size of HNSCC patients in this study. Further investigation with a larger patient cohort is needed to clarify the relationship between ROR1 expression and prognosis in HNSCC. Third, it is possible that the ROR1 peptide binds to HLA-DP or -DQ, in addition to HLA-DR, which may account for the limited suppression of T cell responses observed following anti-HLA-DR antibody treatment. Although we did not perform a pan-class II blockade, the R2 T cell line was established via limiting dilution, suggesting that these cells likely express a single T cell receptor specific for the ROR1_403–417_ peptide presented in the context of HLA-DR. In addition, we assessed the binding affinity of the ROR1_403–417_ peptide to common HLA-DP alleles (DPB102:01 and DPB105:01) and HLA-DQ alleles (DQB103:01, DQB103:03, DQB104:01, and DQB106:01). The peptide demonstrated minimal binding to these HLA-DP and -DQ alleles, indicating that they are unlikely to contribute to the T cell recognition of the ROR1_403–417_. Fourth, ROR1 peptide-specific T cells in circulation were not directly quantified in this study. In our attempt to detect ROR1-specific T cells in peripheral circulation, we initially employed tetramer staining. However, due to the inherent instability of MHC class II tetramers targeting CD4^+^ T cells, the ROR1-specific tetramer was not functional in this study. As an alternative approach, we directly stimulated naïve PBMCs from a healthy control with ROR1 peptides and observed that approximately 0.02% of CD4^+^ T cells responded to the stimulation. This frequency is comparable to that reported for other TAA-reactive CD4^+^ T cells, such as those specific for NY-ESO-1 [47]. Because it was difficult to obtain sufficient blood volume from cancer patients to detect such rare populations directly, we should employ a long-term ex vivo expansion strategy using limited sample volumes in this study. Fifth, the detailed phenotypic characterization of peptide-reactive T cells is lacking in this study. Although the production of IFN-γ and granzyme B in response to the ROR1_403–417_ peptide suggests that the response is primarily mediated by Th1 cells, it remains unclear whether other CD4^+^ T cell subsets also respond to the peptide. Given that cytokine production was comparable between patients and healthy controls in this study, further investigation is warranted to determine whether T cell reactivity is fully preserved in patients. Since the ROR1_403–417_ peptide was designed to bind multiple major HLA-DR alleles (DR1, DR4, DR7, and DR53), which might cover the majority of the population, individuals lacking these alleles may fail to mount a response as suggested. HLA typing should be performed prior to vaccination, even when the epitope is designed to be broadly applicable. Sixth, although this study demonstrated promising immunogenicity, several limitations of peptide-based cancer vaccines targeting TAAs warrant consideration. Vaccine-induced T cells may exhibit limited specificity to tumors and low affinity for naturally processed tumor antigens, thereby mitigating their cytotoxic efficacy. Moreover, the immune response is often restricted to patients possessing specific HLA-DR alleles, which constrains the vaccine’s broad applicability. Insufficient T cell infiltration into the tumor microenvironment may further compromise therapeutic effectiveness. Additionally, immune pressure exerted by vaccination can lead to downregulation of ROR1 expression in tumor cells, facilitating immune evasion. To address these challenges, combining peptide vaccines with immune checkpoint inhibitors, potent adjuvants, or optimized vaccine designs may enhance antitumor efficacy [5].

## 5. Conclusions

This study demonstrated that ROR1 expression is significantly elevated in HNSCC tissues compared to healthy controls, and its high expression is associated with advanced clinical stages. A peptide derived from ROR1 effectively induced CD4^+^ HTLs capable of recognizing and directly killing ROR1+ HNSCC cells. Furthermore, the antitumor efficacy of these ROR1-reactive HTLs was enhanced by ICIs, particularly through the dual blockade of PD-L1/PD-1 and PD-L2/PD-1 axes. These findings highlight the potential of ROR1-targeted peptide vaccines as an effective immunotherapeutic strategy for patients with HNSCC.

## Figures and Tables

**Figure 1 cancers-17-02326-f001:**
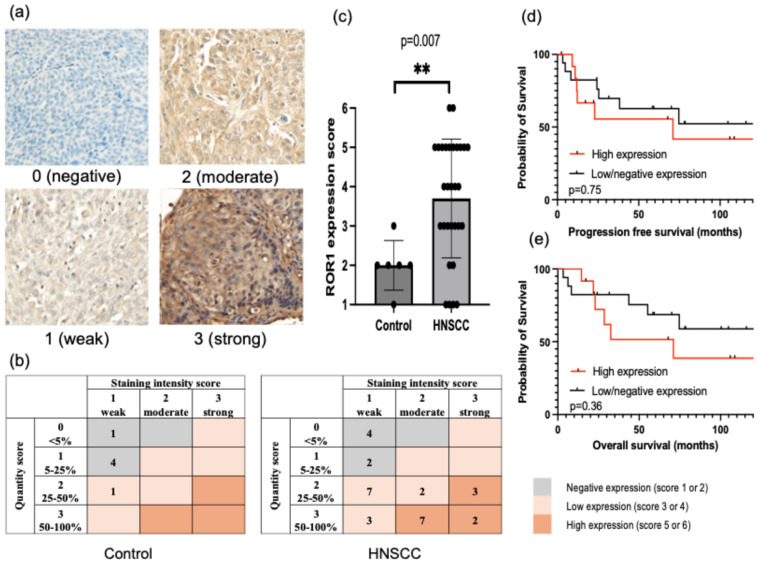
Receptor tyrosine kinase-like orphan receptor 1 (ROR1) expression and clinical relevance in head and neck squamous cell carcinoma (HNSCC). (**a**) Representative immunohistochemistry images of ROR1 in HNSCC tumor tissue, categorized according to staining intensity. (**b**) ROR1 expression scores classified based on staining intensity and the proportion of positive cells. (**c**) Comparison of ROR1 expression scores between healthy controls and patients with HNSCC. (**d**,**e**) Kaplan–Meier curves for progression-free survival (**d**) and overall survival (**e**) in patients stratified by high or low/negative ROR1 expression. Data are presented as median values with 95% confidence intervals. ** *p* < 0.01.

**Figure 2 cancers-17-02326-f002:**
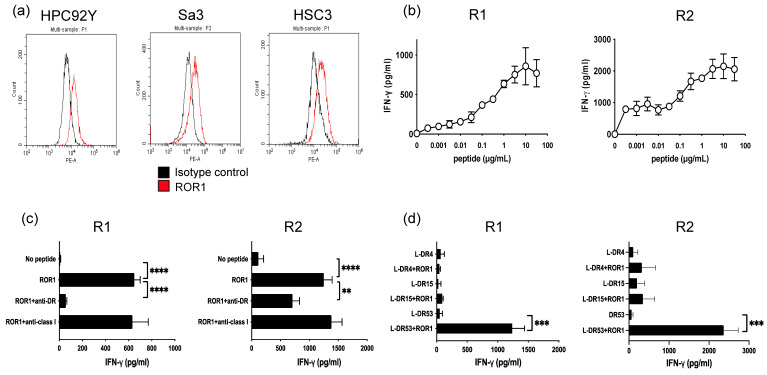
Induction of ROR1-specific T cells by a novel epitope. (**a**) Flow cytometric analysis of ROR1 expression in the HNSCC cell lines HPC92Y, Sa3, and HSC3. (**b**) Evaluation of the peptide dose–responses of ROR1-reactive helper T lymphocytes (HTLs) designated as R1 and R2. These HTLs were co-cultured with autologous peripheral blood mononuclear cells (PBMCs) used as antigen-presenting cells (APCs) and pulsed with increasing concentrations of the ROR1_403–417_ peptide. (**c**) Human leukocyte antigen (HLA) restriction analysis of ROR1-reactive HTLs. These HTLs were cocultured with autologous PBMCs pulsed with 3 μg/mL ROR1_403–417_ peptide with or without anti-HLA-DR or anti-HLA class I antibodies. IFN-γ levels secreted from the co-cultured HTLs were measured using ELISA. (**d**) HLA-DR allele restriction of ROR1-reactive HTLs. HTL responses to the ROR1_403–417_ peptide were evaluated using peptide-pulsed (3 μg/mL) L-cells expressing individual HLA-DR alleles. IFN-γ secretion was quantified via ELISA after 48 h of co-culture. Experiments were performed in triplicate. Data are presented as mean ± standard deviation. ** *p* < 0.01, *** *p* < 0.001, **** *p* < 0.0001.

**Figure 3 cancers-17-02326-f003:**
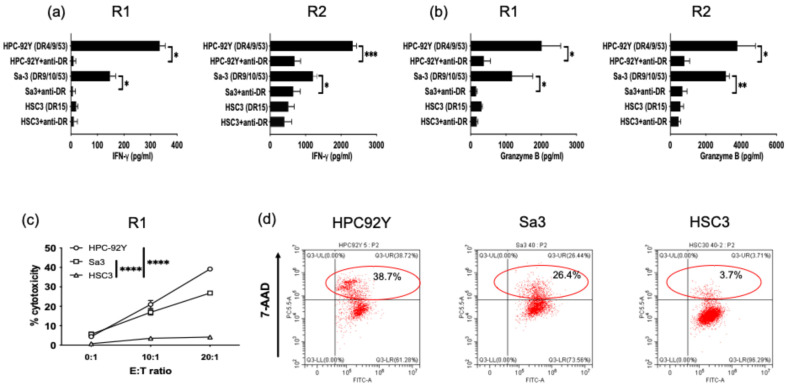
Anti-tumor activity of ROR1-reactive HTLs against ROR1-expressing HNSCC cell lines. (**a**) Direct recognition of tumor cells by ROR1-reactive HTLs was assessed by coculturing HLA-DR-matched or -mismatched HNSCC cell lines with or without anti-HLA-DR antibody. IFN-γ secretion in culture supernatants was measured using ELISA after 48 h of coculture. (**b**) Granzyme B production by ROR1-reactive HTLs under the same conditions was evaluated. (**c**) Cytotoxicity assays of ROR1-reactive HTLs in HLA-DR-matched and -mismatched tumor cell lines. HTLs were co-cultured with tumor cell lines with several effector-to-target (E:T) ratios. (**d**) Representative flow cytometry plots showing tumor cell killing at an E:T ratio of 20:1. Experiments were performed in triplicate (**a**,**b**) or duplicate (**c**,**d**). Data are presented as mean ± standard deviation. * *p* < 0.05, ** *p* < 0.01, *** *p* < 0.001, **** *p* < 0.0001.

**Figure 4 cancers-17-02326-f004:**
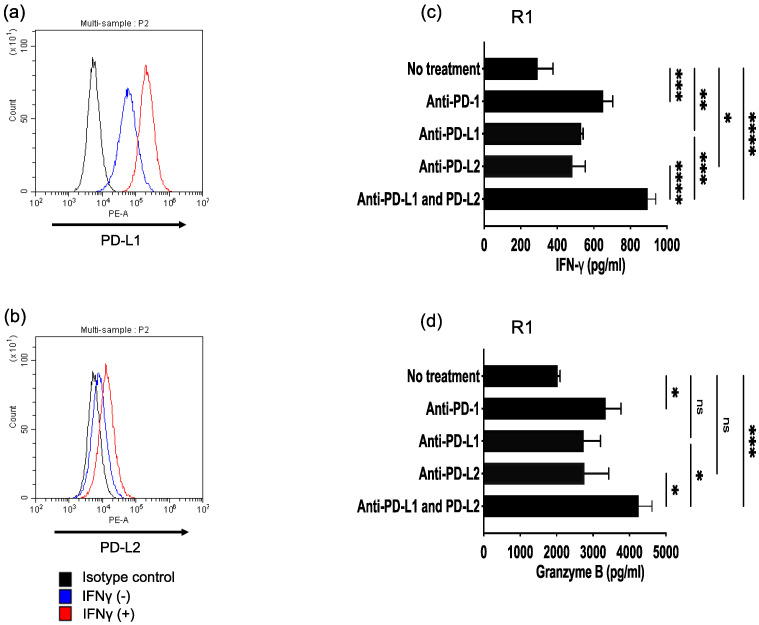
Enhanced antitumor activity of ROR1-reactive HTLs by blockade of PD-L1/PD-1 and PD-L2/PD-1 axes. (**a**,**b**) Expression levels of PD-L1 (**a**) and PD-L2 (**b**) on the HNSCC cell line, HPC92Y, were assessed using flow cytometry. ROR1-reactive HTLs were co-cultured with HPC92Y cells in the presence of immune checkpoint inhibitors targeting PD-1, PD-L1, PD-L2, or a combination of PD-L1 and PD-L2. (**c**,**d**) Levels of IFN-γ (**c**) and granzyme B (**d**) secreted into the supernatants were measured via ELISA after 48 h of coculture. Experiments were performed in triplicate. Data are presented as mean ± standard deviation. * *p* < 0.05, ** *p* < 0.01, *** *p* < 0.001, **** *p* < 0.0001.

**Figure 5 cancers-17-02326-f005:**
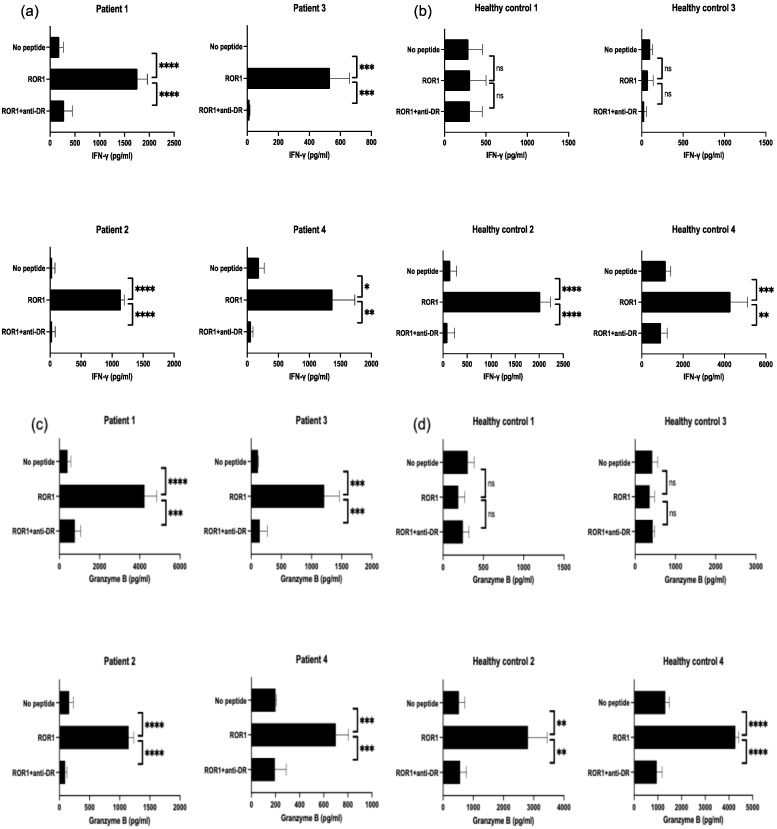
Presence of ROR1-reactive precursor T cells in patients with HNSCC and healthy individuals. (**a**–**d**) PBMCs from patients with HNSCC and healthy individuals were stimulated with the ROR1_403–417_ peptide for two cycles every week. T-cell responses were assessed by measuring IFN-γ (**a**,**b**) and granzyme B (**c**,**d**) levels in culture supernatants. Panels (**a**,**c**) show data from healthy individuals, and panels (**b**,**d**) show data from HNSCC patients. An anti-HLA-DR antibody was used to determine the HLA-DR restriction of the response. Experiments were performed in triplicate. Data are presented as mean ± standard deviation. ns: not significant, * *p* < 0.05, ** *p* < 0.01, *** *p* < 0.001, **** *p* < 0.0001.

**Table 1 cancers-17-02326-t001:** Association between ROR1 expression and clinicopathological features in patients with HNSCC.

	Low/Negative Expression (*n* = 18)	High Expression (*n* = 12)	*p* Value
Age			
<70	11	6	0.71
≧70	7	6	
Sex			
Male	16	12	0.5
Female	2	0	
HPV			
positive	10	4	0.28
negative	8	8	
T classification			
T1–2	13	3	0.02 *
T3–4	5	9	
N classification			
N0–1	12	2	0.01 *
N2–3	6	10	
M classification			
M0	18	12	NA
M1	0	0	
Clinical stage			
I–II	14	4	0.02 *
III–IV	4	8	

* *p* < 0.05. Abbreviations: HPV; human papillomavirus, NA; not applicable, ROR1; receptor tyrosine kinase-like orphan receptor 1.

**Table 2 cancers-17-02326-t002:** Reactivity to the ROR1_403–417_ peptide in PBMCs from patients with HNSCC and healthy controls.

	Sex	Age (Years)	TNM Classification	Clinical Stage	Primary Site	HLA-DR Alleles	IFN-γ (pg/mL)		
							Granzyme B (pg/mL)		
							No Peptide	ROR1	ROR1+ anti-DR
Patient 1	Male	80	T4aN2bM0	IVA	Hypopharyngeal	NA	183 ± 81	1756 ± 201	281 ± 166
							406 ± 156	4234 ± 617	768 ± 293
Patient 2	Female	69	T4aN3bM1	IVC	Hypopharyngeal	NA	31 ± 48	1138 ± 56	32 ± 50
							161 ± 67	1148 ± 84	95 ± 28
Patient 3	Male	53	T2N2M0	III	Nasopharyngeal	NA	<	533 ± 126	14 ± 3
							114 ± 9	1211 ± 248	144 ± 123
Patient 4	Male	69	T4aN0M0	IVA	Laryngeal	NA	186 ± 86	1373 ± 360	59 ± 32
							201 ± 7	700 ± 105	195 ± 95
Healthy control 1	Male	32	NA	NA	NA	4/15/53	288 ± 171	308 ± 195	305 ± 151
							306 ± 78	188 ± 81	247 ± 75
Healthy control 2	Male	31	NA	NA	NA	9/11/53	150 ± 134	2019 ± 216	90 ± 151
							525 ± 190	2810 ± 640	564 ± 207
Healthy control 3	Female	33	NA	NA	NA	12/14	101 ± 27	75 ± 62	29 ± 27
							426 ± 131	360 ± 122	437 ± 48
Healthy control 4	Male	40	NA	NA	NA	9/12/53	1160 ± 246	4284 ± 842	925 ± 308
							1315 ± 169	4274 ± 143	953 ± 230

<: less than lower limit of detection. Abbreviations: IFN-γ; interferon gamma-γ, NA; not applicable/Not Available, ROR1; receptor tyrosine kinase-like orphan receptor 1.

## Data Availability

The data that support the findings of this study are available on request from the corresponding author. The data are not publicly available due to privacy or ethical restrictions.

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
