# Peer review of "ROR1 as an Immunotherapeutic Target for Inducing Antitumor Helper T Cell Responses Against Head and Neck Squamous Cell Carcinoma"

_cancers, 2025, doi:10.3390/cancers17142326_

Round 1
Reviewer 1 Report
Comments and Suggestions for Authors
This manuscript presents a well-designed study exploring ROR1 as a potential immunotherapeutic target in head and neck squamous cell carcinoma (HNSCC). The authors successfully identified a novel ROR1-derived helper T lymphocyte (HTL) epitope (ROR1₄₀₃–₄₁₇) and demonstrated its ability to elicit specific antitumor responses in vitro. The study further reveals that the addition of immune checkpoint inhibitors (ICIs) significantly enhances the cytotoxic activity of ROR1-reactive HTLs.
The research is innovative, methodologically sound, and the manuscript is clearly written. The study adds valuable insight into T cell-based peptide vaccine development, especially in a cancer type with limited responsiveness to ICIs. However, there are several areas where the manuscript could be strengthened to enhance scientific rigor, clarity, and clinical relevance.
My comments are described as follows:
Comments:
- The study focuses on in vitro assays and ex vivo responses. While promising, the lack of in vivo validation such as in animal models or patient-derived xenografts) limits the immediate clinical applicability. The format of tables is not acceptable. Please reorganize the tables. Authors may discuss this limitation more explicitly in the discussion and propose future directions, such as in vivo studies or early-phase clinical trials.
- The study detected ROR1₄₀₃–₄₁₇-reactive T cells in patients with HNSCC and healthy individuals. However, the functional characterization of these patient-derived T cells is limited. Authors may consider including cytokine profile differences or cytotoxicity comparisons between patient-derived vs. healthy donor-derived T cells, or at least acknowledge this as a limitation.
- Consider including cytokine profile differences or cytotoxicity comparisons between patient-derived vs. healthy donor-derived T cells, or at least acknowledge this as a limitation.
- Please expand the discussion with references to expression profiles in critical normal tissues (e.g., lung, heart) and safety profiles from other ROR1-targeted therapies.
- The manuscript mentions no significant differences in OS or PFS between high vs. low/negative ROR1 expression groups, despite a trend. Authors must clarify whether the study is underpowered to detect survival differences or if other factors may be confounding.
Author Response
Reviewer 1
General comments:
The study adds valuable insight into T cell-based peptide vaccine development, especially in a cancer type with limited responsiveness to ICIs.
>Answer: We thank the reviewer for taking time to review our manuscript.
Specific comments:
- The study focuses on in vitro assays and ex vivo responses. While promising, the lack of in vivo validation such as in animal models or patient-derived xenografts) limits the immediate clinical applicability. The format of tables is not acceptable. Please reorganize the tables. Authors may discuss this limitation more explicitly in the discussion and propose future directions, such as in vivo studies or early-phase clinical trials.
>Answer: We agree with the reviewer that the safety and validity of ROR1-targeted immunotherapy should be examined in animal models including patient-derived xenografts. As the safety and efficacy of peptide vaccine using other types of TAAs has been demonstrated both in animal and clinical trials (PMID: 25391695), and we previously reported that a peptide vaccine targeting c-MET exhibited anti-tumor effects in an HNSCC mouse model, we believe that ROR1-targeted immunotherapy may offer clinical benefit in patients with HNSCC. We have incorporated this information into the revised manuscript (Page 14, lines 458–470), and also reorganized the tables as suggested.
- The study detected ROR1₄₀₃–₄₁₇-reactive T cells in patients with HNSCC and healthy individuals. However, the functional characterization of these patient-derived T cells is limited. Authors may consider including cytokine profile differences or cytotoxicity comparisons between patient-derived vs. healthy donor-derived T cells, or at least acknowledge this as a limitation.
>Answer: We additionally evaluated granzyme B production in response to the ROR1403–417 peptide in both HNSCC patients and healthy controls. Similar to IFN-γ, granzyme B responses were observed in all patients with HNSCC and 2 out of 4 healthy controls. These cytokine profiles suggest that the peptide-reactive T cells include, at least in part, a Th1 phenotype. The results of granzyme B have been added to Figure 5c and d, and the limitations have been addressed in the revised manuscript (Page 15, lines 496–503).
- Consider including cytokine profile differences or cytotoxicity comparisons between patient-derived vs. healthy donor-derived T cells, or at least acknowledge this as a limitation.
>Answer: As described above, we have added the data of granzyme B production in response to ROR1403–417 peptide in patients with HNSCC and healthy controls (Figure 5c and d). The production of these cytokines was comparable between patients and healthy controls who responded to the peptide. These results and limitations have been added in the revised manuscript (Page 15, lines 496–503).
- Please expand the discussion with references to expression profiles in critical normal tissues (e.g., lung, heart) and safety profiles from other ROR1-targeted therapies.
>Answer: The expression in healthy tissues is a drawback of TAA. Although Balakrishnan et al. reported positive ROR1 expression in normal adult tissues (parathyroid gland, pancreas, esophagus, stomach, and duodenum), most studies found no or low ROR1 expression in a broad range of normal adult tissues including immune-related organs (bone marrow, peripheral blood immune cells, spleen, thymus, tonsil, and lymph nodes), heart, artery, brain, lung, liver, pancreas, gastrointestinal tract, skeletal muscle, kidney, prostate, testis, ovary, and uterus suggesting that ROR1-targeted therapies have low risk to induce off-target adverse events.
The tolerability of ROR1-targeted therapy in clinical trials further supports the safety of targeting ROR1 as a TAA in peptide vaccine. In a clinical trial using the ROR1-targeted antibody–drug conjugate zilovertamab vedotin, the incidence of adverse events was consistent with other antibody–drug conjugates. Additionally, the safety of ROR1-targeted CAR-T cell has been reported (PMID: 39466024). Nevertheless, the safety and tolerability of ROR1-based peptide vaccines should be carefully evaluated in future studies. We have added this information in the revised manuscript (Page 12, lines 355–364, Page 14, lines 458–464).
- The manuscript mentions no significant differences in OS or PFS between high vs. low/negative ROR1 expression groups, despite a trend. Authors must clarify whether the study is underpowered to detect survival differences or if other factors may be confounding.
>Answer: In this study, high ROR1 expression was associated with advanced clinical stage, T classification, and N classification. Thus, we believe that the lack of statistically significant association between ROR1 and survival is likely due to the relatively small sample size of HNSCC patients in this study (n = 30). Further investigation with a larger patient cohort is needed to clarify the relationship between ROR1 expression and prognosis in HNSCC. We have added this information in the revised manuscript (Page 14, lines 470–476).
Reviewer 2 Report
Comments and Suggestions for Authors
In the manuscript entitled “ROR1 as an immunotherapeutic target for inducing antitumor helper T cell responses against head and neck squamous cell carcinoma”, Sato and co-authors characterize ROR1 tissue expression and CD4 T cell responses against an HLA-DR restricted ROR1 epitope in patients with HNSCC. ROR1 is a known tumor-associated antigen in a range of hematological and solid cancers including HNSCC, and represents a potential therapeutic target. Spontaneous anti-ROR1 immune responses have been described in CLL patients (PMID: 26562161). In clinical trials, anti-ROR1-CAR T cells showed anti-cancer activity in CLL but not HNSCC (PMID: 39466024).
In this study, the authors performed tissue staining on 30 HNSCC tumor samples, finding high ROR1 expression in 40% of samples. Using computational prediction, an HLA-DR53 restricted ROR1 epitope has been identified, followed by the generation of two ROR1-specific CD4 T cell lines from the blood of healthy donors. The two T cell lines displayed reactivity against HLA-DR53 positive targets expressing native antigen or pulsed with ROR1 peptide, as evidenced by interferon production and cell line killing, with responses further enhanced by PD-L1/L2 blockade. Finally, ROR1 peptide-specific responses have been identified in the blood of 4/4 HNSCC patients and 2/4 heathy controls.
The study advances the knowledge of ROR1 as a therapeutic target in HNSCC with potential translational implications for other cancers; regrettably the study did not quantify ROR1 peptide-specific T cells directly in circulation (such as via tetramer or ELISPOT approaches).
The following points need to be addressed or clarified.
- In all figures, please state the number of experimental repeats/biological replicates/technical replicates.
- For all antibodies used in the study, please provide RRIDs and the dilutions used.
- Provide the details of immune checkpoint inhibitors used in Fig 4, and concentrations thereof. Please add the relevant section in the Methods, including whether the cells were pre-treated with IFNγ prior to co-culture with ICIs.
- In Fig 2, ROR1 expression on Sa3 cell line appears negative, and cell surface staining in the other two lines is very low. Is ROR1 differentially distributed? Did the authors perform intracellular staining such as flow or IHC, and/or Western for ROR1 protein quantification?
- In Fig 2c, R2 responses were only moderately reduced by HLADR blockade – is presentation by another allele (or DP, DQ) possible? Did the authors try pan-class II block?
- In Fig 5, negative results in 2/4 healthy controls could be due to the absence of the restricting HLA-DR allele; please complement Table 2 with tissue typing results for HLA-DR allele. Furthermore, the long-term ex vivo expansion approach used is here not quantitative and as such, the results of Fig 5 should not be over-interpreted.
- For Fig 3 c-d, please provide representative examples of flow gating showing identification of tumor cells, T cells, and cell viability.
- In 2.7, was IFNγ washed off prior to testing?
- In the Study Limitations, please consider the following issues pertinent to vaccination: poor specificity; low T cell affinity; reliance on the restricting HLA-DR allele; potentially poor T cell tissue penetration (barriers to tissue trafficking?); loss of tumor ROR1 expression on treatment due to immune pressure. Line 383 mentions melanoma gp100 vaccine +/- IPI – the cited study was published in 2010, and the approach was long since abandoned in favor of ICI whose effect relies on neoantigen-specific, higher affinity T cell responses compared with TAA-specific responses. I suggest expanding the discussion to include these considerations.
Minor
I suggest replacing Fig 1a, b with Supplementary Fig 1 which is more representative of staining variations.
In Fig 1e, please add the p value (**).
Author Response
Reviewer 2
General comments:
The study advances the knowledge of ROR1 as a therapeutic target in HNSCC with potential translational implications for other cancers; regrettably the study did not quantify ROR1 peptide-specific T cells directly in circulation (such as via tetramer or ELISPOT approaches).
>Answer: We thank the reviewer for thoughtful evaluation of our manuscript. In our attempt to detect ROR1-specific T cells in peripheral circulation, we initially employed tetramer staining. However, due to the inherent instability of MHC class II tetramers targeting CD4⁺ T cells, the ROR1-specific tetramer was not functional in this study. As an alternative approach, we directly stimulated naïve PBMCs from a healthy control with ROR1 peptides and observed that approximately 0.02% of CD4⁺ T cells responded to the stimulation. Because it was difficult to obtain sufficient blood volume from cancer patients to detect such rare populations directly, we should employ a long-term ex vivo expansion strategy using limited sample volumes in this study. This information has been incorporated into the revised manuscript (Page 10, lines 317–319, Page 14, lines 486–496).
Specific comments
- In all figures, please state the number of experimental repeats/biological replicates/technical replicates.
>Answer: Details regarding the number of experimental replicates have been included in the revised figure legends.
- For all antibodies used in the study, please provide RRIDs and the dilutions used.
>Answer: We have added the RRIDs and dilutions for all antibodies used in the study to the Methods section.
- Provide the details of immune checkpoint inhibitors used in Fig 4, and concentrations thereof. Please add the relevant section in the Methods, including whether the cells were pre-treated with IFNγ prior to co-culture with ICIs.
>Answer: In figure 4, the HTLs were co-cultured with HPC92Y cells (3 × 10⁴) and anti–PD-1, PD-L1, and PD-L2 antibodies (10 μg/mL) for 48 hours. HPC92Y cells were pretreated with IFN-γ (500 IU/mL) for 48 hours and thoroughly washed before further analysis. Details regarding the use of immune checkpoint inhibitors and IFN-γ pretreatment have been incorporated into the Methods section (Page 4, lines 164–168).
- In Fig 2, ROR1 expression on Sa3 cell line appears negative, and cell surface staining in the other two lines is very low. Is ROR1 differentially distributed? Did the authors perform intracellular staining such as flow or IHC, and/or Western for ROR1 protein quantification?
>Answer: We re-evaluated the expression of cell surface ROR1 in Sa3 cells using flow cytometry and confirmed its positivity (Figure 2a). As suggested, we additionally conducted western blot and immunocytochemical analysis of ROR1 in HNSCC cell lines. In addition to membrane localization, ROR1 was also detected in the cytoplasm and nucleus, consistent with previous findings by Tseng et al. (PMID: 20577074). No apparent differences in expression were observed among the tumor cells. These results have been incorporated into Supplemental Figure 1 and the revised manuscript (Page 7, lines 230–233).
- In Fig 2c, R2 responses were only moderately reduced by HLADR blockade – is presentation by another allele (or DP, DQ) possible? Did the authors try pan-class II block?
>Answer: We thank the reviewer for the thorough evaluation of our manuscript. Although we did not perform a pan-class II blockade, the R2 T cell line was established via limiting dilution, suggesting that these cells likely express a single T cell receptor specific for the ROR1403–417 peptide presented in the context of HLA-DR. In addition, we assessed the binding affinity of the ROR1403–417 peptide to common HLA-DP alleles (DPB102:01 and DPB105:01) and HLA-DQ alleles (DQB103:01, DQB103:03, DQB104:01, and DQB106:01). The peptide demonstrated minimal binding to these HLA-DP and -DQ alleles, indicating that they are unlikely to contribute to T cell recognition of the ROR1403–417. This information has been included in the revised manuscript (Page 14, lines 477–486).
- In Fig 5, negative results in 2/4 healthy controls could be due to the absence of the restricting HLA-DR allele; please complement Table 2 with tissue typing results for HLA-DR allele. Furthermore, the long-term ex vivo expansion approach used is here not quantitative and as such, the results of Fig 5 should not be over-interpreted.
>Answer: We have added the HLA-DR allele information of healthy controls in Table 2. Among the two individuals who did not respond to the ROR1403–417 peptide, one possessed an HLA-DR allele predicted to bind the peptide, while the other carried an allele that may not. Due to the limited availability of patient blood samples, HLA typing could not be performed in this cohort. Since the ROR1403–417 peptide was designed to bind multiple major HLA-DR alleles (DR1, DR4, DR7, and DR53), which might cover the majority of the population, individuals lacking these alleles may fail to mount a response as suggested. Ideally, HLA typing should be performed prior to vaccination, even when the epitope is designed to be broadly applicable. This point has been addressed in the revised Discussion section (Page 15, lines 503–507).
We agree with the reviewer that long-term ex vivo expansion may not accurately reflect the in vivo frequency of TAA-reactive T cells. To assess the presence of ROR1-specific T cells in peripheral circulation, we initially employed tetramer staining. However, due to the inherent instability of MHC class II tetramers targeting CD4⁺ T cells, the ROR1-specific tetramer was not functional in this study. As an alternative, we stimulated naïve PBMCs from healthy control 2 with ROR1 peptides and observed that approximately 0.02% of CD4⁺ T cells responded to peptide. This frequency is comparable to that reported for other TAA-reactive CD4⁺ T cells, such as those specific for NY-ESO-1 (PMID: 33637530). Because it was difficult to obtain sufficient blood volume from cancer patients to detect such rare populations directly, we employed a long-term ex vivo expansion strategy using limited sample volumes. We have included data showing direct detection of IFN-γ–producing CD4⁺ T cells by flow cytometry following short-term stimulation with the ROR1403–417 peptide (Supplemental Figure 3). We thank the reviewer for raising these important points and have included a discussion on the limitations of the long-term ex vivo expansion approach in the revised manuscript (Page 10, lines 317–319, Page 14, lines 486–496).
- For Fig 3 c-d, please provide representative examples of flow gating showing identification of tumor cells, T cells, and cell viability.
>Answer: Representative examples of flow cytometry gating strategies used to identify tumor cells and assess cell viability have been added to Supplemental Figure 2 (Page 8, lines 271–272).
- In 2.7, was IFNγ washed off prior to testing?
>Answer: IFN-γ was thoroughly washed out prior to subsequent analysis. We have added this information to the Method section (Page 4, lines 164–165).
- In the Study Limitations, please consider the following issues pertinent to vaccination: poor specificity; low T cell affinity; reliance on the restricting HLA-DR allele; potentially poor T cell tissue penetration (barriers to tissue trafficking?); loss of tumor ROR1 expression on treatment due to immune pressure.
>Answer: We thank the reviewer for insightful suggestions to improve the quality of the discussion. As suggested, we have addressed the limitations of TAA-targeted peptide vaccine therapy including poor specificity, low T cell affinity, reliance on the restricting HLA-DR allele, potentially poor T cell tissue penetration, and loss of tumor ROR1 expression due to immune pressure in the Discussion section (Page 145, lines 507–517).
- Line 383 mentions melanoma gp100 vaccine +/- IPI – the cited study was published in 2010, and the approach was long since abandoned in favor of ICI whose effect relies on neoantigen-specific, higher affinity T cell responses compared with TAA-specific responses. I suggest expanding the discussion to include these considerations.
>Answer: We agree with the reviewer that ICIs partly rely on neoantigens, which exhibit significantly higher antigenicity compared to TAAs. ICIs would be more likely to exert their effects via neoantigens rather than TAAs. However, neoantigens can be lost, particularly in tumors with high mutational burden, leading to immune evasion. Moreover, the identification of neoantigens is both costly and time-consuming. While the functional potency of TAA-specific T cells is generally lower than that of neoantigen-specific T cells, we believe that exploring the therapeutic potential of TAA-specific T cells in combination with ICIs remains a compelling area of investigation. This point has been incorporated into the revised Discussion section (Page 13, lines 418–436).
- I suggest replacing Fig 1a, b with Supplementary Fig 1 which is more representative of staining variations.
>Answer: We have replaced Figure 1a and 1b with Supplementary Figure 1 as suggested.
- In Fig 1c, please add the p value (**).
>Answer: We have added the p value in Figure 1c.
Round 2
Reviewer 1 Report
Comments and Suggestions for Authors
Thank you for authors revsion. I have no more questions.